# Potential Risk of Other-Cause Mortality Due to Long-Term Androgen Deprivation Therapy in Elderly Patients with Clinically Localized Prostate Cancer Treated with Radiotherapy—A Confirmation Study

**DOI:** 10.3390/jcm9072296

**Published:** 2020-07-20

**Authors:** Hideya Yamazaki, Koji Masui, Gen Suzuki, Norihiro Aibe, Daisuke Shimizu, Takuya Kimoto, Ken Yoshida, Satoaki Nakamura

**Affiliations:** 1Department of Radiology, Urology, Graduate School of Medical Science, Kyoto Prefectural University of Medicine, Kyoto 602-8566, Japan; mc0515kj@koto.kpu-m.ac.jp (K.M.); gensuzu@koto.kpu-m.ac.jp (G.S.); a-ib-n24@koto.kpu-m.ac.jp (N.A.); dshimizu@koto.kpu-m.ac.jp (D.S.); t-kimoto@koto.kpu-m.ac.jp (T.K.); 2Department of Radiology, Kansai Medical University, Hirakata 573-1010, Japan; yoshidaisbt@gmail.com (K.Y.); satoaki@nakamura.pro (S.N.)

**Keywords:** prostate cancer, androgen deprivation therapy, elderly, OCM, HDR

## Abstract

Androgen deprivation therapy (ADT) is used to improve overall survival (OS) in prostate cancer treatment; however, we encountered that long-term ADT in elderly patients may be related to high other-cause mortality (OCM). This study aimed to confirm the potential risk associated with long-term ADT in elderly patients using a different large cohort. A comparison analysis was conducted between the ≥2- and <2-year ADT groups using open, large data from 1840 patients with clinically localized prostate cancer treated with radiotherapy (1172 treated with high-dose-rate brachytherapy (HDR) + external beam radiotherapy (EBRT) and 668 treated with external beam radiotherapy). The OCM-free survival (OCMFS), overall survival, and prostate cancer-specific survival rates were measured. The 10-year OCMFS rates in patients aged ≥75 years were 94.6% and 86% in the <2- and ≥2-year ADT groups, respectively, but were 96.3% and 93.5% (*p* = 0.0006) in their younger counterparts. If dividing into HDR and EBRT groups. This inclination was found in brachytherapy group but not in EBRT group. The overall survival rate was also lower in the elderly patients in the ≥2-year ADT group than in the <2-year ADT group; however, the 10-year prostate cancer-specific survival rate was the same in both groups. Long-term ADT in elderly patients resulted in not only higher OCM rates but also poorer OS rates; therefore, longer-term ADT in elderly patients should be performed with meticulous care.

## 1. Introduction

In 2018, an estimated 1.3 million new cases of prostate cancer and 359,000 associated deaths occurred worldwide, making it the second most frequent cancer and fifth leading cause of cancer death in men [1]. Androgen deprivation therapy (ADT) is used to improve prognosis, and its efficacy in combination with standard radiation therapy up to 70 Gy has been confirmed by several randomized controlled trials (RCTs) [2,3]. However, in modern radiotherapy, wherein higher doses of ≥74 Gy can be delivered, there is little evidence to support the role of additional ADT [3,4]. Furthermore, as ADT causes several untoward effects such as diabetes and cardiovascular toxicity, physicians should employ meticulous caution while treating fragile patients [5,6], including the elderly, to prevent other-cause mortality (OCM), i.e., death due to causes other than prostate cancer [5,6]. Recently, the outcomes of prostate cancer treatment have improved, and a nearly 100% prostate cancer-specific relative survival rate has been achieved [3,7]; therefore, the simultaneous importance of OCM has increased. Previously, we unexpectedly found that long-term ADT ≥ 2 years increases the risk of OCM in patients aged ≥75 years in a study on 1125 patients with localized prostate cancer treated with modern high-dose RT including low-dose-rate brachytherapy [8]. The 10-year OCM-free survival (OCMFS) rates in patients aged ≥75 years were 86.8% and 60.7% in the <2- and ≥2-year ADT groups, respectively, whereas they were 90.0% and 86.8% (*p* < 0.0001) in their younger counterparts. Furthermore, overall survival (OS) was also poorest in elderly patients who received ADT for ≥2 years [8]. This unexpected previous finding prompted us to conducted a confirmation study using different databases open for public use that included >1800 patients with clinically localized prostate cancer treated with RT including high does rate brachytherapy (HDR) [9]. Therefore, this study aimed to examine the influence of long-term ADT, according to age, on survival after radiotherapy.

## 2. Methods

### 2.1. Patients

We conducted a comparative study on multi-institutional retrospective study data open for public use (B17-278) [9]. A total of 1901 patients with histology-proven prostate adenocarcinoma treated with HDR and external beam radiotherapy (EBRT) or EBRT alone with a curative intent were screened for eligibility based on the following criteria: node negative cancer, metastasis free status, availability and accessibility of data for identifying OCM and ADT details, and a minimum 1-year follow-up for surviving patients or until death for non-survivors. Subsequently, the final study included 1840 patients (1172 treated with HDR + EBRT and 668 treated with EBRT). Cancer was staged according to the National Comprehensive Cancer Network (NCCN) 2015 risk classification as follows [3]: low risk, T1–T2a, Gleason score (GS) 2–6, or pretreatment prostate-specific antigen (initial PSA) level < 10 ng/mL; intermediate risk, T2b–T2c, GS 7, or PSA level of 10–20 ng/mL; and high risk, T3a–T4, GS 8–10, or PSA level >20 ng/mL.3 PSA failure was defined using the Phoenix definition (nadir + 2 ng/mL). The median follow-up for the entire cohort was 66 months (ranging from 2 to 177 months), with a minimum of 1 year for surviving patients or until death for non-survivors. This study was conducted in accordance with the Declaration of Helsinki.

### 2.2. Treatment

The detailed method of applicator implantation in HDR was described elsewhere [10]. A total of 1172 patients were treated with a combination of HDR and EBRT at various fractionations (Table 1). The median dose of HDR was 31.5 Gy (11–31.5 Gy), and that of EBRT was 39 Gy (39–45 Gy). The median fraction size of HDR was 9 Gy (6.4–9.0 Gy), and that of EBRT was 30 Gy (30–51 Gy). The EBRT group consisted of 1152 patients who received three-dimensional conformal RT (3D-CRT) and 20 who received intensity modulated RT (IMRT). Regarding ADT, almost all patients received (94%) neoadjuvant ADT, and 817 patients (88.4%) received adjuvant ADT. Of the 668 patients in the EBRT group, 105 received 2D + 3D CRT, 240 received 3D-CRT, and 299 received IMRT. The median dose of EBRT was 72 Gy (62–80 Gy) in 36 (20–40) fractions.

### 2.3. Statistical Analysis

StatView 5.0 statistical software was used for the statistical analyses. Percentages were analyzed using the chi-square test, and the Student’s *t*-test was used for normally distributed data. The Mann–Whitney U test for skewed data was used to compare means or medians. The Kaplan–Meier method was used to analyze survival, and comparisons were made using the log-rank test. Cox’s proportional hazards model was used to calculate hazard risk. A *p*-value < 0.05 was considered statistically significant.

## 3. Results

### 3.1. Patients and Treatment Characteristics

Of the 1840 patients, 1076 received ADT for >2 years, and 764 received it for <2 years. The median age was 70 years (47–89 years). Basic characteristics of the patients and their treatment are presented in Table 1. A comparison of the background characteristics between the <2- and ≥2-year ADT groups is shown in Table 1. The ≥2-year ADT group included patients with an advanced disease (higher T category, higher initial PSA level, higher Gleason score sum, and higher risk group in the NCCN risk classification) treated more frequently with HDR + EBRT than with EBRT alone.

### 3.2. Other-Cause Mortality (OCM), Prostate Cancer-Related Death, and OS

OCM occurred in 63 patients (Table 2).

Basic characteristics of aged patients were depicted in Table 3.

The 10-year OCMFS rates in patients aged ≥75 years were 94.6% (95% confidence interval [CI], 89.4–99.7%) and 86% (95% CI, 77.7–94.3%) in the <2- and ≥2-year ADT groups, respectively, whereas they were 95.2% (95% CI, 93.9–98.6%) and 93.5% (95% CI, 90.7–96.3%) in their younger counterparts (Figure 1a, *p* = 0.0006 among 4 groups, *p* = 0.0442 for comparison between <2- and ≥2-year ADT groups in elderly, and *p* = 0.2853 in young counterpart). Hazard ratio of OCMSF in patients aged ≥75 years were 1.433 (95% CI, 0.510–4.025; *p* = 0.4947) and 3.823 (95% CI, 1.852–7.889, *p* = 0.0003) in the <2- and ≥2-year ADT groups, respectively, whereas they were 1 (reference group) and 1.427 (95% CI, 0.739–2.757; *p* = 0.2891) in their younger counterparts.

The 10-year OS rates in the elderly group were 91.7% (85.3–98.1%) and 84.7% (76.3–93.1%) in the <2- and ≥2-year ADT groups, respectively, whereas they were 91.4% (87.8–94.9%) and 90.8% (87.5–94.1%) in the younger patients (Figure 1b, *p* = 0.0108 among 4 groups, *p* = 0.0473 for comparison between <2- and ≥2-year ADT groups in elderly, and *p* = 0.9233 in young counterparts). The hazard ratio of OS in patients aged ≥75 years were 0.992 (95% CI, 0.431–2.280, *p* = 0.9844) and 2.246 (95% CI, 1.256–4.015; *p* = 0.0064) in the <2- and ≥2-year ADT groups, respectively, whereas they were 1 (reference group) and 0.975 (95% CI, 0.598–1.589; *p* = 0.9189) in their younger counterparts.

The 10-year prostate cancer-specific survival rates in patients aged ≥75 years were 97.0% and 98.5% in the <2- and ≥2-year ADT groups, respectively, but they were 94.9% and 97.1% (Figure 1c, *p* = 0.4958) in their younger counterparts. Long-term ADT in elderly patients influenced not only OCM but also OS. 

If divided by the duration of ADT, a statistically significant difference was observed in total and elderly, and intermediate risk group (Table 4).

### 3.3. Comparison between EBRT and HDR + EBRT Group

We made a comparison between EBRT and HDR + EBRT group. Table 5 shows background comparison between two groups. 

In EBRT group, the 10-year OCMFS rates in patients aged ≥75 years were 94.8% and 100% in the <2- and ≥2-year ADT groups, respectively, whereas they were 96.3% and 83.6% in their younger counterparts (Figure 2a, *p* = 0.373 among 4 groups, not available for comparison between <2- and ≥2-year ADT groups in elderly, and *p* = 0.1253 in young counterpart). In HDR-BT group, The 10-year OCMFS rates in patients aged ≥75 years were 92.9% and 84.2% in the <2- and ≥2-year ADT groups, respectively, whereas they were 96.0% and 94.7% in their younger counterparts (Figure 2a, *p* = 0.0002 among 4 groups, *p* = 0.9525 for comparison between <2- and ≥2-year ADT groups in elderly, and *p* = 0.3835 in young counterparts).

### 3.4. Causes of OCM

Table 6 shows the causes of OCM. Other malignancies were a major cause of OCM. In the <2-year ADT group, no cardiovascular death was recorded, whereas two patients died of cardiovascular disease in the ≥2-year ADT group. The unknown causes included sudden death.

## 4. Discussion

We present evidence that ADT does not always improve survival outcomes after RT for patients with localized prostate cancer. In fact, long-term ADT for ≥2 years has a negative effect on OCM and OS, especially in elderly patients aged ≥75 years [8], which was reported in a previous cohort and confirmed in this study. The strength of this study is that large cohorts were used, including >1000 patients, in both a previous experimental arm [8] and this confirmation arm.

Generally, ADT has been recognized as an important intervention in the management of prostate cancer. Huggins and Hodges [11,12] initialized a successful ADT strategy in the early 1940s in which castration arrested the growth of prostate cancer cells and suppressed serum prostate phosphatases in metastatic prostate cancer cells. Several RCTs confirmed that simultaneous RT with ADT is a useful treatment for high-risk prostate cancer with RT doses of up to 70 Gy [2,3,4]. Modern technical advancement in RT enabled us to increase the irradiation dose to the target area without administering unnecessary higher doses to the surrounding normal tissue, which improved the outcome of localized prostate cancer RT; modern high-dose RT (e.g., IMRT and BT) combined with ADT is recognized as the standard treatment for locally advanced and/or intermediate- to high-risk disease [3,4]. However, there is little evidence to support these combinations of higher-dose RT ≥ 74 Gy and/or BT. Furthermore, the optimum duration of ADT with higher-dose RT is yet to be determined [3,4,13].

It is often said that most men with prostate cancer will die with the disease rather than of the disease because the prostate cancer-specific survival rate has improved [1,3,7,8]. Our data confirmed this tenet, because most men died of causes other than prostate cancer, and long-term ADT ≥2 years worsened not only OCM but also OS. Several adverse effects of ADT have been reported: 5 weight gain, decreased muscle mass and increased insulin resistance, decreased bone mineral density, decreased libido and sexual dysfunction, hot flashes, gynecomastia, reduced testicle size, anemia, fatigue, diabetes, cardiovascular events (myocardial infarction [MI] and sudden cardiac death) [14,15], cerebrovascular diseases [15], kidney injury [16], dementia [17], and thromboembolic events [18,19]. Abdollah et al., in their study including 137,524 patients with non-metastatic prostate cancer treated between 1995 and 2009, reported that treatment with medical ADT may increase the risk of OCM [6]. Morgans et al. cited the importance of age because they speculated that the risk of incident diabetes mellitus (DM) or cardiovascular disease in men exposed to long-term ADT for ≥2 years increases with age at diagnosis, especially in those ≥75 years [20]. In contrast, most studies, including a meta-analysis of RCTs, reported that ADT is not related to greater cardiovascular mortality [3,5].

Several studies describe the negative effects of ADT after RT [21,22,23]. Beyer et al. found a poorer 10-year OS rate that decreased from 44% in hormone-naïve patients receiving low dose rate BT (LDR-BT) to 20% in patients receiving ADT and under treatment with LDR-BT, with the leading causes of death being cardiovascular death and prostate and other cancers [21]. This finding was confirmed by other studies. Nanda reported that neoadjuvant ADT is significantly associated with an increased risk of all-cause mortality among men with a history of coronary artery disease (CAD), CAD-induced chronic heart failure, or MI who are treated with LDR-BT but not among men with no comorbidity or a single CAD risk factor [22]. Pickles et al. also confirmed increased cardiovascular death rates after LDR-BT in intermediate-risk patients treated with ADT [23]. Our findings are in line with these results. Especially, in comparison between EBRT and brachytherapy, we found that negative effect of long-term ADT was only found in HDR-BT group, and the strength of this study is that we examined the effect of ADT on OCM in patients treated with modern RT with a high biologically effective dose, focusing on age.

In this decade, only few patients with localized prostate cancer died of prostate cancer. Therefore, we consider this an important finding not only for radiation oncologists but also for general healthcare practitioners. Whether this is a simple association or a cause–effect relationship is unknown. At least, our data raise a question to longer-term ADT use in elderly patients without evidence, which should be performed with meticulous care.

This study remains several limitations. The primary limitation of this study is its retrospective design and inherent selection bias associated with treatment being as rendered. A longer follow-up with a larger number of homogenous patients is needed before establishing concrete conclusions. Further, a comorbidity analysis was lacking, including geriatric assessment (i.e., Geriatric 8 Score proposed by SIOP), and this is important because DM and cardiovascular diseases are confirmed important factors influencing OCM [8,14,15,20,22]. Additionally, the lack of serum testosterone measurements could be a concern since many older men will have a prolonged recovery time after 2 years of ADT. Therefore, this study could not examine the role of preventive measures or targeted intervention. Finally, the reason for OCM could not be specified; it is not always the cardiovascular system and is mainly other cancers. The hypothesis explains the mechanism of unexpected OCM is required to mitigate toxicity.

In conclusion, long-term ADT for ≥2 years is correlated with a risk of mortality due to causes other than prostate cancer, in patients with localized prostate cancer aged ≥75 years.

## Figures and Tables

**Figure 1 jcm-09-02296-f001:**
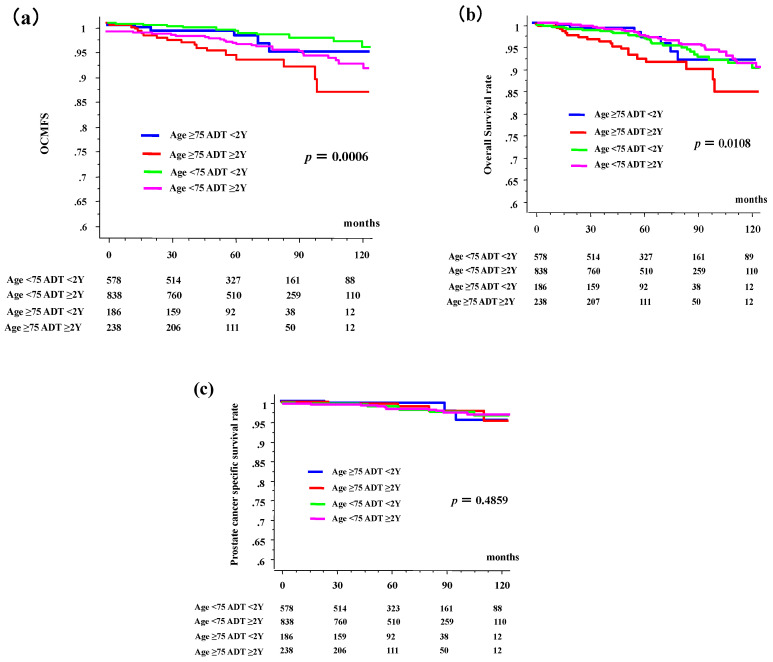
Other-cause of mortality (OCM), overall survival (OS), prostate cancer-specific survival, according to duration of androgen deprivation therapy (ADT) and age. (**a**) OCM-free survival rate (OCMFS). (**b**) Overall survival rate (OS). (**c**) Prostate cancer-specific survival rate. The time = 0 represents the time of start of radiotherapy.

**Figure 2 jcm-09-02296-f002:**
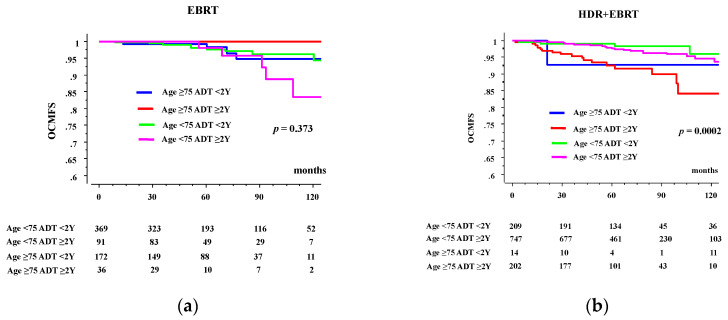
OCM according to duration of ADT and age. (**a**) OCMFS in external beam radiotherapy (EBRT) group. (**b**) OCMFS in high dose rate brachytherapy plus EBRT (HDR + EBRT) group.

**Table 1 jcm-09-02296-t001:** Characteristics and treatment factors of patients according to androgen deprivation therapy (ADT).

Variables	Strata	ADT ≥ 2 Years		ADT < 2 Years		*p*-Value
		*n* = 1076		*n* = 764		
		No. or Median (range)	(%)	No. or Median (range)	(%)	
Age	50	59	(5%)	43	(6%)	0.4271
	60	434	(40%)	280	(37%)	
	70	345	(32%)	255	(33%)	
	75-	238	(22%)	186	(24%)	
T category	1	188	(17%)	190	(25%)	**<0.0001**
	2	340	(32%)	286	(37%)	**exc NA**
	3	529	(49%)	267	(35%)	
	4	11	(1%)	18	(2%)	
	NA	8	(1%)	3	(0%)	
Pretreatment PSA	ng/mL	16.68 (2.682–1454)		12.4 (2.286–399)		**<0.0001**
Gleason score	−6	100	(9%)	70	(9%)	**<0.0001**
	7	462	(43%)	451	(59%)	**exc NA**
	8-	510	(47%)	243	(32%)	
	NA	4	(0.4%)	0	(0%)	
NCCN risk classification	Low	7	(1%)	26	(3%)	**<0.0001**
	Intermediate	139	(13%)	271	(35%)	**exc NA**
	High	927	(86%)	467	(61%)	
	NA	3	(0.3%)	0	(0%)	
Modality	EBRT	127	(12%)	541	(71%)	**<0.0001**
	HDR + EBRT	949	(88%)	223	(29%)	
Follow-up	Months	68 (2–177)		63 (4–165)		0.1135

Bold values indicate statistically significance. NA: not available. HDR: high-dose-rate brachytherapy. EBRT: external beam radiotherapy. exc NA = *p*-vale was calculated excluding NA.

**Table 2 jcm-09-02296-t002:** Characteristics and treatment factors for OCM.

Variables	Strata	OCM (+)		OCM (-)	*p*-Value
		*n* = 63		*n* = 1777	
		No. or Median (range)	OCM (+) %	No. or Median (range)	
Age	−59	2	(2%)	100	**0.0043**
	60–69	12	(2%)	702	
	70–74	27	(5%)	573	
	75–	22	(5%)	402	
T category	1	13	(3%)	365	**0.1869**
	2	15	(2%)	611	**exc NA**
	3	34	(4%)	762	
	4	0	(0%)	29	
	NA	1	(9%)	10	
Pretreatment PSA	ng/mL	19.45 (3.30–329)		15.072 (2.286–1454)	0.2069
Gleason	≤6	4	(2%)	166	0.1075
score	7	24	(3%)	889	exc NA
	8≤	33	(4%)	720	
	NA	2		2	
NCCN risk	Low	0	(0%)	33	**0.0209**
classification	Intermediate	12	(3%)	398	excl. NA
	High	50	(4%)	1344	
	NA	1		2	
PSA failure	Yes	12	(5%)	231	0.1835
	No	51	(3%)	1550	
	NA			1	
ADT	No	1	(1%)	145	0.0979
	Yes	62	(4%)	1637	
	ADT <2-y	17	(3%)	606	
	ADT ≥2-y	45	(4%)	1031	
	Duration	40 (0–113)		35.5 (0–49)	**0.0096**
Neoadjuvant	(yes)	62		1625	
(duration)	Months	10 (4–55)		9 (1–92)	
Adjuvant	(yes)	50		1209	
(duration)	Months	36 (10–50)		36 (1–134)	
Modality	EBRT	18	(3%)	655	0.1847
	HDR + EBRT	45	(4%)	1127	

Bold values indicate statistically significance. NA: not available. HDR: high-dose-rate brachytherapy. EBRT: external beam radiotherapy. exc NA = *p*-vale was calculated excluding NA.

**Table 3 jcm-09-02296-t003:** Characteristics and treatment factors of elder patients and younger counterpart.

Variables	Strata	Elder ≥ 75		Young < 75		*p*-Value
		*n* = 424		*n* = 1416		
		No. or Median (range)	(%)	No. or Median (range)	(%)	
T category	1	87	(21%)	291	(21%)	0.3333
	2	149	(35%)	477	(34%)	
	3	183	(43%)	613	(43%)	
	4	2	(0.5%)	27	(2%)	
	NA	3	(0.7%)	8	(1%)	
Pretreatment PSA	ng/mL	15.039 (2.682–500)		15.155 (2.286–1454)		0.5979
Gleason score	≤6	39	(9%)	131	(9%)	0.5485
	7	204	(48%)	709	(50%)	
	8≤	179	(42%)	574	(41%)	
	NA	2	(0.5%)	2	(0.1%)	
NCCN risk	Low	6	(1%)	27	(2%)	**0.0028**
classification	Intermediate	80	(19%)	330	(23%)	
	High	335	(79%)	1059	(75%)	
	NA	3	(1%)	0	(0%)	
ADT	Yes	399	(94%)	1296	(92%)	0.0839
	No	25	(6%)	120	(9%)	
	Duration	36 (0–102)		33 (0–140)		0.7073
Neoadjuvant	(yes)	62		1625		
(duration)	Months	10 (4–55)		9 (1–92)		
Adjuvant	(yes)	50		1209		
(duration)	Months	36 (10–50)		36 (1–134)		
Follow-up	Months	58.5 (2–155)		68 (4–177)		**<0.0001**
Modality	EBRT	208	(49%)	460	(33%)	**<0.0001**
	HDR + EBRT	216	(51%)	956	(68%)	

Bold values indicate statistically significance. NA: not available. HDR: high-dose-rate brachytherapy. EBRT: external beam radiotherapy.

**Table 4 jcm-09-02296-t004:** Characteristics and treatment factors for OCM according to ADT duration.

Variables	Strata	ADT Duration	OCM (+)		OCM (-)	*p*-Value
			*n* = 63		*n* = 1777	
			No.	OCM (+) %	No.	
Total		ADT < 2-y	18	(3%)	602	**0.0338**
		ADT ≥ 2-y	45	(4%)	1031	
NCCN risk classification	Low	ADT < 2-y	0	(0%)	26	NA
		ADT ≥ 2-y	0	(0%)	7	
	Intermediate	ADT < 2-y	4	(1%)	267	**0.0337**
		ADT ≥ 2-y	8	(6%)	131	
	High	ADT < 2-y	14	(3%)	453	0.4013
		ADT ≥ 2-y	36	(4%)	891	
Age	Elder ≥ 75	ADT < 2-y	5	(3%)	181	**0.0401**
		ADT ≥ 2-y	17	(7%)	221	
	Young < 75	ADT < 2-y	13	(2%)	565	0.2283
		ADT ≥ 2-y	28	(3%)	810	

Bold values indicate statistically significance.

**Table 5 jcm-09-02296-t005:** Comparison of background between EBRT and HDR + EBRT group.

Variables	Strata	EBRT		HDR + EBRT		*p*-Value
		*n* = 668		*n* = 1172		
		No. or Median (range)	(%)	No. or Median (range)	(%)	
Age	50	25	(4%)	77	(7%)	**<0.0001**
	60	213	(32%)	501	(43%)	
	70	222	(33%)	378	(32%)	
	75-	208	(31%)	216	(18%)	
T category	1	140	(21%)	238	(20%)	**0.0257**
	2	225	(33%)	401	(34%)	exc NA
	3	279	(41%)	517	(44%)	
	4	19	(3%)	19	(2%)	
	NA	5	(1%)	5	(0%)	
Pretreatment PSA	ng/mL	16.2 (2.28–1454)		14.7 (2.68–500)		**<0.0001**
Gleason score	≤6	68	(10%)	102	(9%)	**0.0222**
	7	357	(53%)	556	(47%)	exc NA
	8≤	242	(36%)	511	(44%)	
	NA	1	(0%)	3	(0%)	
NCCN risk classification	Low	22	(3%)	11	(1%)	**<0.0001**
	Intermediate	174	(26%)	236	(20%)	exc NA
	High	472	(70%)	922	(79%)	
	NA	0	(0%)	3	(0%)	
PSA failure	Yes	148	(22%)	95	(8%)	**<0.0001**
	No	520	(77%)	1076	(92%)	exc NA
	NA	0	(0%)	1	(0%)	
ADT	Yes	578	(86%)	1117	(95%)	**<0.0001**
	No	90	(13%)	55	(5%)	
	Duration	9 (0–140)		43 (0–128)		**0.0274**
OCM	Yes	18	(3%)	45	(4%)	0.194
	No	650	(97%)	1127	(96%)	
Follow-up	Months	61.0 (9–145)		69 (9–177)		**0.0033**

Bold values indicate statistically significance. NA: not available. HDR: high-dose-rate brachytherapy. EBRT: external beam radiotherapy. exc NA = *p*-vale was calculated excluding NA. HDR + EBRT groups included patients with younger, advanced disease with more ADT, and lower PSA failure rate.

**Table 6 jcm-09-02296-t006:** Cause of other cause of mortality (OCM).

Total	ADT ≥ 2Y		ADT < 2Y	
	*n* = 1076		*n* = 764	
Cardiovascular	2	(0.2%)	0	(0.0%)
Pulmonary	4	(0.4%)	2	(0.3%)
Other malignancies	15	(1.4%)	2	(0.3%)
Other	6	(0.6%)	4	(0.5%)
Unknown	18	(1.7%)	10	(1.3%)
Elder ≥ 75	*n* = 238		*n* = 186	
Cardiovascular	0	(0.0%)	0	(0.0%)
Pulmonary	2	(0.8%)	0	(0.0%)
Other malignancies	8	(3.4%)	1	(0.5%)
Other	1	(0.4%)	1	(0.5%)
Unknown	6	(2.5%)	3	(1.6%)
Young < 75	*n* = 838		*n* = 578	
Cardiovascular	2	(0.2%)	0	(0.0%)
Pulmonary	2	(0.2%)	2	(0.3%)
Other malignancies	7	(0.8%)	1	(0.2%)
Other	5	(0.6%)	3	(0.5%)
Unknown	12	(1.4%)	7	(1.2%)

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
