# Peer review of "Potential Risk of Other-Cause Mortality Due to Long-Term Androgen Deprivation Therapy in Elderly Patients with Clinically Localized Prostate Cancer Treated with Radiotherapy—A Confirmation Study"

_jcm, 2020, doi:10.3390/jcm9072296_

Round 1

Reviewer 1 Report

Minor, typos

Line 43 : 5,6

Line 52 : prompted us to conduct….

Line 54: .9

Line 119 : elder or elderly (see title)

Line 118 : what do you mean by aged patients

Line 119 : <75 years of age is not really „young“ please reconsider this lable

Line 170 : 2,3,4

Line 175 : 3,4

Line 181: :5

Line 190: 20

Line 214: older/elder or elderly

Line 130 /134: younger counterparts or young counterparts: I prefer younger because < 75 y is not really young.

Major:

  • The reader wants to know more precisely how long patients received ADT in both groups. (Mean duration, Max.- Min.)
  • The findings are interesting and important. Besides the already mentioned limitations of the study the authors should stress more clearly that study design was not only retrospective but that there were signifcant differences between both ttt arms (ADT > 2y and ADT < 2 y) e.g. use of brachytherapy. This further limits the study.
  • Furthermore the study is based on numeric age in a geriatric population. Was a geriatric assessement peformed prior to ttt in patients > 75 years as recommended by the SIOG? G8 Score for example? If not this should als be mentioned as a limitation of the study and included into the recommendation.

Author Response

Response to reviewer
Thanks for your suggestions to improve our manuscript.

Reviewer 1.

Minor, typos

Line 43 : 5,6

We deleted them accordingly.

Line 52 : prompted us to conduct….

We corrected accordingly.

Line 54: .9

We corrected accordingly.

Line 119 : elder or elderly (see title)

We changed “elder” and “young” to  “Age ≥75” and “Age <75”.

Line 118 : what do you mean by aged patients

We changed word “aged” to “elderly”.

Line 119 : <75 years of age is not really „young“ please reconsider this lable

We changed “young” to “younger”.

Line 170 : 2,3,4

We corrected accordingly.

Line 175 : 3,4

We corrected accordingly.

Line 181: :5

We corrected accordingly.

Line 190: 20

We corrected accordingly.

Line 214: older/elder or elderly

We changed to “elderly”.

Line 130 /134: younger counterparts or young counterparts: I prefer younger because < 75 y is not really young.

We used “younger counterparts” according to you suggestion.

Major:

The reader wants to know more precisely how long patients received ADT in both groups. (Mean duration, Max.- Min.)

We added details of ADT (median duration, min., max) and we added a detailed period of administration into neoadjuvant and adjuvant period in addition to total periods in Table 1 and 2.

The findings are interesting and important. Besides the already mentioned limitations of the study the authors should stress more clearly that study design was not only retrospective but that there were significant differences between both ttt arms (ADT > 2y and ADT < 2 y) e.g. use of brachytherapy. This further limits the study.

Thanks a lot for your thoughtful suggestion.  Until now, we did not think that there is difference between BT and EBRT.   However, we are surprised to find that the correlation (long-term ADT and high prevalence of OCM in elderly) was only found in BT group but not in EBRT group in this dataset.  So that we added a section “comparison between EBRT and BT” in result and added a phrase in discussion.

In addition, we recognized that all preceding studies citing risk of ADT on radiotherapy were using brachytherapy (low dose rate brachytherapy=LDR), which has a good coincidence to our dataset, is also surprising us and gave us a new insight into this phenomenon.  It may be because radiotherapy with curative intensity (=brachytherapy) eradicated cancer cells and reduced prostate cancer specific mortality, which enhanced the role of other cause of mortality.  We inserted a phrase “Especially, in comparison between EBRT and brachytherapy, we found that negative effect of long-term ADT was only found in HDR-BT group, and all previous reports also arise from brachytherapy series.  On possibility is that brachytherapy has an enough higher intensity to control prostate cancer, so that the role of OCM would be enhanced.” in discussion.

Again, thanks to your important suggestion to deepen our study.

Furthermore, the study is based on numeric age in a geriatric population. Was a geriatric assessement peformed prior to ttt in patients > 75 years as recommended by the SIOG? G8 Score for example? If not this should als be mention

Thanks to your thoughtful suggestion. In general, radiotherapy (especially brachytherapy) was performed for patients with good general status (ECOG performance status 0-1). However databased we examined did not contain the information for general status.  Therefore, I added a sentence” including geriatric assessment (i. e., : Geriatric 8 Score)” in limitation of discussion.

Reviewer 2 Report

The writing is poor. Introduction is basically a rehash of the abstract.

Not clear whether the <2 year ADT group had terminated their ADT, or the ADT was ongoing. What other treatments were they undergoing at the time?

Table 5 – the text accompanying Table 2 states “In the <2-year ADT group, no cardiovascular death was recorded, whereas two patients died of cardiovascular disease in the ≥2-year ADT group”. Table 5 actually shows that in the <2-year ADT group, 2 cardiovascular deaths were recorded, whereas no patients died of cardiovascular disease in the ≥2-year ADT group. Please check!

In Figure 1, what does the time = 0 represent? Is that the time of ADT initiation?

Not clear from the text or tables the times of RT relative to ADT.

The conclusion that “long-term ADT for ≥2 years is correlated with a risk of mortality due to causes other than prostate cancer, in patients with localized prostate cancer aged ≥75 years” has been shown by others. The new angle in this manuscript is the correlation with RT. Is there any difference between EBRT and EBRT+HDR?

The study is not suitable for a mortality study because the number of patients who died is relatively low.

Finally, although the conclusion corroborates that if other studies, the data actually does not support this hypothesis. The number of patients who died is too small. There is no correlation between ADT and a specific co morbidity – although the correlation between ADT and heart disease has been established in other studies, this current study fails to show that correlation. No correlation between ADT and pulmonary disease has been shown. If there is one, it would be interesting. Similarly, there is currently no known correlation between ADT and other cancers. If the authors can show this correlation, that will be a novel finding. However, that evidence is not present in the current study.

Round 2

Reviewer 1 Report

Changes were made according to my suggestions. I have nor more changes.